# In Search of Dispersed Memories: Generative Diffusion Models Are Associative Memory Networks

**DOI:** 10.3390/e26050381

**Published:** 2024-04-29

**Authors:** Luca Ambrogioni

**Affiliations:** Donders Institute for Brain, Cognition, and Behaviour, Radboud University, 6525 XZ Nijmegen, The Netherlands; luca.ambrogioni@donders.ru.nl

**Keywords:** generative diffusion models, associative memory networks, hopfield networks

## Abstract

Uncovering the mechanisms behind long-term memory is one of the most fascinating open problems in neuroscience and artificial intelligence. Artificial associative memory networks have been used to formalize important aspects of biological memory. Generative diffusion models are a type of generative machine learning techniques that have shown great performance in many tasks. Similar to associative memory systems, these networks define a dynamical system that converges to a set of target states. In this work, we show that generative diffusion models can be interpreted as energy-based models and that, when trained on discrete patterns, their energy function is (asymptotically) identical to that of modern Hopfield networks. This equivalence allows us to interpret the supervised training of diffusion models as a synaptic learning process that encodes the associative dynamics of a modern Hopfield network in the weight structure of a deep neural network. Leveraging this connection, we formulate a generalized framework for understanding the formation of long-term memory, where creative generation and memory recall can be seen as parts of a unified continuum.

## 1. Introduction

Memory is a mysterious thing. Human beings can form lifelong memories from fleeting events and effortlessly recall them decades later as vivid multi-sensory experiences. On the other hand, in spite of their impressive capabilities, deep learning systems tend to require extensive training sessions to encode new information, which severely limit their adaptivity and consequently their capacity to develop general intelligence. Nevertheless, the field of machine learning has a long history of research on biologically plausible memory [1,2,3,4,5,6]. Arising from the pioneering work of John Hopfield, associative memory networks have been proposed as computational models of biological memory [1,7]. These networks encode memories as stable fixed-points in an “energy landscape” defined on the space of neural activations. Interestingly, this energy function can be encoded into a pattern of synaptic connections trained using biologically plausible synaptic learning rules [2,8]. In recent years, associative memory networks have been generalized in order to greatly scale their encoding capacity [9,10]. However, these modern Hopfield networks have a much more tenuous connection with known forms of synaptic learning since their energy function cannot be straightforwardly captured by learned pairwise synaptic couplings [11]. While biologically plausible implementations of modern Hopfield networks have been proposed, they do not provide strong insights on how memories are encoded in synaptic patterns since they require hard storage of the memorized pattern [11].

While human-like long-term memory is still outside of the capabilities of artificial intelligence systems, great progress has been made in approximating several forms of human creativity. Modern generative models are nowadays capable of generating beautiful visual art and insightful written text. In fact, these generative models have attracted great attention in neuroscience since they can provide an internal model of the world [12,13,14] and form the mechanistic basis for imagination and top-down predictive perception [15]. Interestingly, modern research in psychology and neuroscience suggests that there is no sharp distinction between memory and spontaneous imagination [16]. In particular, the theory of reconstructive memory suggests that most sensory aspects of our memories are not encoded but are instead reconstructed from contextual information [16,17,18,19]. Memory-related brain areas such as the hippocampus have been shown to respond to imagination, future prediction and counterfactual reasoning tasks [16]. Moreover, hippocampal replays can be seen as a form of spontaneous generation, which is considered to be vital for memory consolidation and planning [20]. Given these deep connections, it is natural to expect that generative models and associative memory models are two faces of the same coin.

In this paper, we will show that this is indeed the case for generative diffusion models, a relatively new class of generative models that have achieved state-of-the-art performance in most computer vision and audio generation tasks [21,22,23]. Specifically, we show that the asymptotic low-time energy landscape of a large class of generative diffusion models trained on discrete patterns is identical to the asymptotic low-temperature landscape of modern Hopfield networks. Furthermore, in our experiments, we show that this equivalence holds almost exactly in the standard numerical implementations.

Using these results, we offer a new theoretical conceptualization of associative memory that can incorporate semantic, episodic, and reconstructive memory as the result of the action of the same synaptic learning rule.

## 2. Preliminaries

In this section, we will review the basic theory of associative memory and diffusion modeling. To keep the text readable to a large audience, we will focus on the intuitive aspects and keep the level of mathematical formalism at a minimum. For example, we will write stochastic differential equations in terms of (infinitesimal) update equations. For a more formal treatment of these topics, we recommend the reader to refer to SDE texts such as to Kloeden et al. [24].

### 2.1. Associative Memory Networks

Hopfield networks have been developed to formalize the concept of associative memory in a simplified artificial neural system [1]. We will denote the activity of the *D* memory units with a vector $x\left(t\right)=(x_{1}\left(t\right),\cdots ,x_{D}\left(t\right))$. In the original paper, these neural activities where assumed to be binary variables ($x_{j}\left(t\right)\in \left\{-1,1\right\}$), respectively, denoting states of rest and states of firing. The dynamic of a Hopfield network is regulated by the update equation:(1)$$x_{j}\left(t+dt\right)={\left[\mathrm{sign}\left(Wx\right)\right]}_{j},$$
where *W* is a real-valued symmetric matrix of synaptic couplings (weights) with null diagonal ($W_{jj}=0,\;\forall j$). The matrix *W* encodes pairwise associations between the different components of the pattern vectors. It can be show that this update rule decreases monotonically the following energy function [1]:(2)$$u_{H}\left(x\right)=x^{T}Wx,$$
and that it will, therefore, converge to one of its local minima. These minima can then be used to encode memories, which can be retrieved through the Hopfield dynamics when initialized in an incomplete or perturbed version of the memory state. The simplest way to encode memories in the coupling matrix is to use the Hebb’s rule of association, which in its simplest form is:(3)$$W_{j,k}=\sum_{n=1}^{N}y_{n,j}y_{n,k}$$
where the set of vectors $\left\{y_{1},\cdots ,y_{n},\cdots y_{N}\right\}$ represents *N* “experienced” patterns of neural activity. A pattern is considered to be successfully stored if it is a stable fixed-point of the discrete dynamics. If the patterns are random, it can be proven that the storage capacity of Hopfield nets scales as $D/4\log_{2}D$ [7].

The storage capacity of Hopfield-like networks can be increased by including non-linear mappings F(·) in its energy function [9,10,25]. The general form for the energy of a (discrete) modern Hopfield network can be written as:(4)$$u_{F}\left(x\right)=h\left(\sum_{n=1}^{N}F\left(x^{T}y_{n}\right)\right),$$
where h(·) is an arbitrary differentiable and strictly monotonic function, which does not affect the location and stability of the local minima. This expression reduces to the standard Hopfield energy for $F\left(x\right)=x^{2}$ and h(x)=x. However, it is possible to achieve much higher theoretical capacity by using more complex functions. For example, a modern Hopfield network with $F\left(x\right)=e^{x}$ can store up to $2^{D/2}$ binary patterns [10]. These associative networks have been recently generalized to have continuous dynamics. If the activation vector is continuous, the energy function needs to include a regularization term to enforce stability. For example, [26] proposed the use of an energy function of the form:(5)$$u_{\mathrm{MH}}\left(x,\beta \right)=-\beta^{-1}\log\left(\sum_{n=1}^{N}e^{\beta x^{T}y_{n}}\right)+{∥x∥}_{2}^{2}/2$$
where β is a positive-valued parameter. We omitted the terms that are additive constant in x since they do not change the fixed-points and the resulting dynamics. In these models, the variable x is assumed to be real-valued and the resulting energy-minimization dynamics can be expressed either through update rule or through a system of ordinary differential equations. Moreover, recent work [11] showed that this dynamics can be expressed in terms of biologically plausible binary association between latent neurons, in a way that is similar to the architecture of a restricted Boltzmann machine [27]. Unfortunately, in these models, the numerical values of the patterns directly determine the synaptic weights between latent and observable neurons. This implies that the patterns need to be stored in memory instead of being converted into distributed synaptic patterns. In this sense, modern Hopfield networks offer a model of memory recall but do not provide insight into learning and memory storage in the brain.

### 2.2. Generative Diffusion Models

Consider a target distribution ϕ(y) that is only available through a training dataset $D=\{y_{1},\cdots ,y_{N}\}$ of independently sampled data-points. Our goal is to learn the structure of the training set in order to generate new samples from ϕ(y). To this aim, we first define a noise-injective process that turns the training samples into random noise states. We will then ‘invert’ this process to turn random noise into new samples. To be consistent with the notation used in the continuous Hopfield model, we deviate from the diffusion modeling literature by writing this process in reversed time, with the noiseless data corresponding to a final time *T*. In this notation, the diffusive dynamics can be determined by the following backward recursive Equation (corresponding to the forward process described in the standard diffusion literature):(6)$$x\left(t-dt\right)=x\left(t\right)+\sigma \sqrt{dt}\delta \left(t\right),$$
where σ determines the standard deviation of the noise injected at time *t* and δ(t) follows a standard normal distribution. This is known as the variance exploding equation in the generative diffusion modeling literature [23], which corresponds to a mathematical Brownian motion. We can sample from this process by sampling an initial state y from the dataset and using it as the final state in the recursion defined by Equation (Equation 6). Note that any other Itô diffusion process defined by a stochastic differential equation can equivalently be used to define a generative diffusion model [23], we will cover the general case in Section 3.2. It can be shown that, if Equation (Equation 6) is initialized with the target distribution ϕ(y), the ‘inverse’ equation is given by:(7)$$x\left(t+dt\right)=x\left(t\right)+\sigma^{2}\nabla_{x}\log p_{t}\left(x\left(t\right)\right)dt+\sigma \sqrt{dt}\delta \left(t\right).$$
where $p_{t}\left(x\right)$ is the marginal distribution of the noise-injection process at time *t*. In the case of the variance-exploding process, the marginal can be computed analytically and it can be expressed as: (8)$$p_{t}\left(x\right)=E_{y∼\phi (y)}\;\left[\frac{1}{\sqrt{2\pi \left(T-t\right)\sigma^{2}}}e^{-\frac{{∥x-y∥}_{2}^{2}}{2\left(T-t\right)\sigma^{2}}}\right].$$

This formula involves an average over the distribution of the data, which is usually not available in a generative modeling task. However, we can approximate the drift term of the dynamics (i.e., the so-called score $\nabla_{x}\log p_{t}(x)$) using a parameterized deep network s(x(t),t;W) trained using the denoising loss [23,28,29]:(9)$$L\left(W\right)=\frac{1}{2}E_{y∼D,t}\;\left[E_{x(t)|y}\;\left[{∥\delta (t)-s(x(t),t;W)∥}_{2}^{2}\right]\right]$$
where δ(t)=(x(t)−y)/(σt) is the total noise added to the pattern y up to time *t*, and *t* is uniformly sampled over [0,T]. The score can then be recovered from the network using the formula:(10)$$\nabla_{x}\log p_{t}\left(x\right)\approx -\sigma^{-1}s\left(x\left(t\right),t;W\right)$$
which becomes exact when the dataset is infinitely large and the loss is minimized globally.

## 3. The Equivalence between Diffusion Models and Modern Hopfield Networks

We can now show that, when used for storing discrete patterns, the dynamics of generative diffusion models minimizes the energy function of continuous modern Hopfield networks. We will start from the simpler case of variance-exploding diffusion models as their analysis provides all the main insights and results. We will then generalize the result to arbitrary diffusion models defined by a large class of SDEs.

### 3.1. Variance-Exploding Models

The first step is to formulate the deterministic dynamics of the model as the negative gradient of an energy function: $x\left(t+dt\right)=x\left(t\right)-\nabla_{x}u\left(x,t\right)dt+\sigma \sqrt{dt}\delta \left(t\right).$ As noted in [30], the energy function is:(11)$$u_{\mathrm{DM}}\left(x,t\right)=-\sigma^{2}\log p_{t}\left(x\right)=-\sigma^{2}\log E_{y∼\phi (y)}\;\left[e^{-\frac{{∥x-y∥}_{2}^{2}}{2\left(T-t\right)\sigma^{2}}}\right]+c,$$
where *c* does not depend on x and can, therefore, be omitted without affecting the dynamics. In order to establish a link between the diffusion model and Hopfield networks, we can now assume that the data source is a finite collection of *N* patterns that we wish to store as memories. This led to the energy:(12)$$u_{\mathrm{DM}}\left(x,t\right)=-\sigma^{2}\log\left(\frac{1}{N}\sum_{n=1}^{N}e^{-\frac{{∥x-y_{n}∥}_{2}^{2}}{2\left(T-t\right)\sigma^{2}}}\right).$$

If we now assume that the patterns are normalized (${∥y∥}_{2}^{2}=1$), by expanding the square in the exponent and omitting constant additive terms, we obtain:(13)$$u_{\mathrm{DM}}\left(x,t\right)/\sigma^{2}=-\log\left(\sum_{n=1}^{N}e^{\frac{x^{T}y_{n}}{\left(T-t\right)\sigma^{2}}}\right)+\frac{{∥x∥}_{2}^{2}}{{\left(T-t\right)}^{2}}.$$

Finally, if we define $\beta {\left(t\right)}^{-1}=\left(T-t\right)\cdot \sigma^{2}$ and we multiply both sides by $\beta {\left(t\right)}^{-1}$, we obtain:(14)$$\beta^{-1}\left(t\right)u_{\mathrm{DM}}\left(x,t\right)/\sigma^{2}=-\beta {\left(t\right)}^{-1}\log\left(\sum_{n=1}^{N}e^{\beta \left(t\right)x^{T}y_{n}}\right)+\frac{{∥x∥}_{2}^{2}}{2}=u_{\mathrm{MH}}\left(x,\beta \left(t\right)\right),$$
which for a fixed *t* is identical to the continuous Hopfield network energy in Equation (Equation 5), and it, therefore, has the same fixed-point structure at the limit β→∞. Note that the scaling factor $\beta^{-1}\left(t\right)/\sigma^{2}$ does not change the fixed-points and their stability as it is a positive constant of x. While we derived this result assumes normalized patterns, this is not actually necessary since, for β(t)→∞, we have that $\beta^{-1}\left(t\right)u_{\mathrm{DM}}\left(x,t\right)/\sigma^{2}∼-\frac{1}{2}\left(x^{T}y^{*}+{∥y^{*}∥}_{2}^{2}+{∥x∥}_{2}^{2}\right)$, where $y^{*}$ is the pattern that maximizes the quadratic form $x^{T}y+{∥y∥}_{2}^{2}$. As shown by this expression, at this limit the norm $y^{*}$ only adds an irrelevant constant shift in the energy.

The main differences between the two approaches are (1) in diffusion models, β(t) tends toward this divergent limit as part of the denoising dynamics, while the denoising iterations of modern Hopfield networks keep β fixed and (2) in Hopfield networks, the energy function is minimized deterministically, whereas in diffusion models, there is an additional stochastic term. However, these two differences ‘cancel each other out’ since the divergence of β(t) leads to the suppression of the stochastic fluctuations and to exact convergence on the same patterns that minimize the modern Hopfield energy for β→0. As shown in our experiments, there is no meaningful difference as far as β is large enough.

### 3.2. The General Case

We can now prove the equivalence in a much more general case where the noise injection dynamics follows the equation:(15)$$x\left(t-dt\right)=x\left(t\right)+\nabla v\left(x\right)dt+\sigma \left(t\right)\sqrt{dt}\delta \left(t\right),$$
where v(x) is a differentiable scalar potential function and σ(t) is a continuous function of time. This form covers most of the equations used in the diffusion modeling literature but it excludes non-conservative dynamics and state-dependent noise models. The generative dynamics corresponding to this more general noise-injection model is given by the equation:(16)$$x\left(t+dt\right)=x\left(t\right)+\left(\sigma {\left(t\right)}^{2}\nabla_{x}\log p_{t}\left(x\left(t\right)\right)-\nabla v\left(x\right)\right)dt+\sigma \left(t\right)\sqrt{dt}\delta \left(t\right).$$
where the conditional marginal distributions of the stochastic differential equation is given by the expression:(17)$$p_{t}\left(x\right)=E_{y∼\phi (y)}\;\left[k(x(t),t;y,T)\right].$$

In this formula, the solution kernel $k(x^{\prime \prime },t^{\prime \prime };x^{\prime },t^{\prime })$ gives the conditional probability density of a state $x^{\prime }$ at $t^{\prime }$ to be moved to $x^{\prime \prime }$ at time $t^{\prime \prime }$ under the noise-injecting dynamics. Unfortunately, this solution kernel cannot be expressed analytically in the general case. We define the function ψ(x,t;y)=logk(x,t;y,T) as the logarithm of the solution kernel, which allows us to express the energy function of the model in the following form:(18)$$u_{v\mathrm{DM}}\left(x,t\right)=-\sigma^{2}\left(t\right)\log\left(\sum_{n=1}^{N}e^{\psi (x,t;\;y_{n})}\right)+v\left(x\right)\;$$

As we showed in the previous section, the equivalence with the Hopfield model can be shown at the limit t→T, which corresponds to β→∞. At this limit, the diffusion dynamics of Equation (Equation 17) around a pattern $y^{*}$ is well-approximated by the linearized equation:(19)$$x\left(t-dt\right)-x\left(t\right)=\nabla v\left(y*\right)dt+H\left(y*\right)\left(x-y^{*}\right)dt+\sigma \left(t\right)\sqrt{dt}\delta \left(t\right),$$
with H(y*) denoting the Hessian matrix of the potential at y*, with $H_{ij}\left(y*\right)=\frac{\partial^{2}u\left(y^{*}\right)}{\partial x_{j}\partial x_{j}}$. Since the equation is linear, its solution kernel is given by a Gaussian density whose mean vector and covariance matrix depend on the potential v(x) and on σ(t). For t→T, the drift term is negligible compared to the diffusion term as the former scales linearly while the latter scales as a squared root. For this reason, the kernel simplifies further into the following asymptotic expression:(20)$$\psi \left(x,t;y\right)∼-\frac{1}{2\sigma^{2}\left(T\right)\left(T-t\right)}{∥x-y∥}_{2}^{2}+c.$$
where *c* does not depend on x. Up to multiplicative and additive constants, this leads to the asymptotic energy function:(21)$$u_{v\mathrm{DM}}\left(x,t\right)∼-\log\left(\sum_{n=1}^{N}e^{-\frac{{∥x-y_{n}∥}_{2}^{2}}{2\left(T-t\right)\sigma {\left(T\right)}^{2}}}\right),$$
which is identical to the energy function in Equation (Equation 12). Note that we neglected the “regularization term” v(x), since it does not diverge for t→T and it is, therefore, negligible at this limit. From this result, we can conclude that the fixed-point structure of the diffusion model at t→T does not depend on the specific form of the SDE and it is therefore fully characterized by the Brownian motion model analyzed in Section 3.1.

## 4. Encoding Memories by Denoising Neural State

To date, we considered exact diffusion models by analyzing the theoretically optimal score function. However, in real-world applications it is often not possible to compute the score analytically. Instead, as outlined in Section 2.2, the score function is approximated using a denoising deep neural network s(x(t),t;W) parameterized by a large array *W* of synaptic weights. Therefore, in real applications the memories are ultimately encoded in the pattern of synaptic weights updated by SGD. Specifically, for a single data-point, the loss function in Equation (Equation 9) results to a SGD weight updates of the form:(22)$$W_{t+1}=W_{t}-\eta \frac{\partial s}{\partial W}\left(\delta (t)-s(x,t;W)\right).$$
where δ(t) can be interpreted as the residual between a past low-noise state y and a current noisy-state x. In the brain, we conjecture that the residual might be obtained through very short-term sensory memory, which can buffer recent states and then compare them with the current activity. In psychology and neuroscience, this form of ‘buffering’ is often referred to as iconic memory, which is estimated to last approximately 1000 ms [31,32,33]. Interestingly, recent studies show that the backpropagation term ∂s/∂W can also be obtained through noise injection [34], which suggests that noise-injecting processes may play a major role in learning. While the activity of the brain is ongoing and cannot be neatly separated into forward and reversed dynamics, there is tantalizing evidence that the time-evolution of cortical networks oscillate between feed-forward and feed-back phases [35], which corresponds to states with different signal-to-noise ratios [36]. This phenomenon might offer a mechanistic implementation of generative diffusion training in the brain, possibly in relation with the hippocampus theta cycle, which is known to play a crucial role in memory consolidation [37]. Coordination between the frontal cortex is likely to play a central role in the consolidation of memory traces [38,39,39] and in the modulation of the oscillatory cycles [40,41]. This suggests that a denoising training might be carried out through frontal-hippocampal feedback process, where cycles of forward and reversed processes are alternated in order to learn both from real and simulated experiences (replays). However, these are speculative suggestions that need to be tested through detailed biophysical theoretical modeling and experimental studies.

## 5. Beyond Classical Associative Memories

In the previous section, we showed that the asymptotic energy landscape of generative diffusion models trained on a finite set of discrete patterns matches the energy function of modern Hopfield networks for β→∞. This implies that, in this regime, the two models have the same fixed-point structure and, consequently, that they have the same memory capacity. Nevertheless, generative diffusion models are more general than associative memory networks in two respects: (1) they provide a framework for probabilistic memory recall and (2) they can be used to encode non-discrete memory structures of arbitrary dimensionality. In this sense, generative diffusion model theory can be seen as a wide generalization of the classical theory of associative memory.

### 5.1. Probabilistic Recall

At its core, the classical theory of associative memory is centered around the idea of deterministic energy minimization (however, see [42] for related probabilistic methods). On the other hand, the theory of generative diffusion modeling is based on partially random processes. This is not a coincidence since these models have been devised for the purpose of generation and statistical variability is central to the generative task. However, probability theory is also central to the problem of memory retrieval, since in general there are many possible patterns y that could have generated a given corrupted/incomplete state x. This is particularly central when the level of corruption is high (i.e., for large values of (T−t)), since in this regime it is often impossible to distinguish between several possible patterns and the best hope is to cover their (posterior) probability distribution. When initialized in one of these partially corrupted states, the stochastic generative dynamics of diffusion models can be interpreted in this fashion, with each denoising trajectory eventually reaching one of the possible fixed-points that are compatible with the initial state. Probabilistic recall can be used to encode Bayesian uncertainty, with several possible memories being simultaneously co-activated by parallel denoising processes. This form of encoding can be used to solve information gathering, exploration and other meta-learning problems [43] and has been suggested to be central to the functioning of the mammalian hippocampus [44].

### 5.2. Higher-Dimensional Memory Structures

In the classical associative memory literature, a memory is encoded as a single (fixed-) point in a *m*-dimensional space of possible neural activities. In modern mathematics, a set of points is often interpreted as a 0-dimensional space as there are no local degrees of freedom, meaning that it is impossible to locally perturb a point while remaining inside its ‘space’. The energy landscape and associated vector field of this form of memories is visualized in Figure 1a. This presents a very strict definition of memory that is incompatible with its cognitive reality in most biological and artificial forms of intelligence. In fact, even the sharpest human memory only encodes a minority of the details of the original neural state, with many of these details being reconstructed during recall based on contextual information [19]. This suggests that a more realistic description of memory should include “internal” degrees of freedom that allow for partial encoding. In mathematical terms, this can be done by defining a memory as a connected *d*-dimensional sub-space embedded in the larger *m*-dimensional space of possible neural activities. A diffusion model trained on *N* of these “extended memories”, each represented by a sub-space $S_{n}$ has the energy function:(23)$$\log\left(\sum_{n=1}^{N}\int_{S_{n}}e^{\beta \left(t\right)x^{T}y_{n}}dy_{n}\right),$$
where the integral is taken over the whole sub-space $S_{n}$, whose points correspond equally valid interpretations of the same memory. An example of an energy landscape for this form of extended localized memory is given in Figure 1b. In general, each sub-space $S_{n}$ can potentially have a different dimensionality or even be a more complex geometrical structure with fractional (fractal) or variable dimensionality.

## 6. A Theoretical Framework for Memory in Biological and Artificial Intelligent Systems

In this section, we will leverage the connection we established between associative memory and diffusion models in order to outline a theoretical framework for biologically plausible memory. Our goal is both to provide conceptual tools to theoretical and experimental neuroscientists and to promote developments in naturalistic machine intelligence systems.

### 6.1. Semantic, Episodic, and Reconstructive Memory

In psychology and neuroscience, the semantic (or structural) memory system is thought to learn the general structure of the sensory input, discarding the idiosyncratic details of individual events. This can be modeled with an energy function where the sum over patterns is replaced by an integral over a continuous density ϕ(y):(24)$$\sigma^{2}\log\int e^{\beta \left(t\right)x^{T}y}\phi \left(y\right)dy.$$

In practice, the distribution ϕ(y) is often defined on a manifold or some other lower dimensional structure, leading to dynamics similar to what is visualized in Figure 1c. This is exactly the kind of behavior we expect from a generative model, initial perturbed states are gradually pushed towards a point on the manifold of possible patterns. In a sense, it can also be seen as a probabilistic generalization of the high dimensional memory structures described in Equation (Equation 23), where the sub-space $S_{n}$ is no longer localized in a small sub-region of the space of neural states. In cognitive terms, roughly speaking, we can say that a memory is “episodic” if its sub-space is localized, whereas a memory is considered semantic or structural if it is part of a large, non-localized sub-space.

However, modern memory research suggests that real-life episodic memories are thought to be largely reconstructive, meaning that most of the sensory details are re-created during recall based on contextual information [18,45]. For example, the memory of a car crash may evoke the memory of broken glass, although the windshield was not actually broken during the real event. This suggests that human episodic memory has a stored lower-dimensional “representational core” that does not fully constrains the dynamics of the system. This can be formalized using a mixture of discrete (or localized) and continuous distributions:(25)$$\log\left(\sum_{n}^{N}e^{\beta \left(t\right)f_{n}{\left(x\right)}^{T}\xi_{n}}+\int e^{-\beta \left(t\right)x^{T}y}\phi \left(y\right)dy\right),$$
where the function $f_{n}:R^{D}\to R^{W}$ with W<D is a lower dimensional encoding of the state and $\xi^{n}\in R^{W}$ is a stored lower-dimensional pattern that encode the “core” of the memory. Since W<D, this energy will only constrain *W* degrees of freedom to converge to the pattern, while the other degrees of freedom are left free to evolve under the dynamics determined by the corpus of semantic memory. This leads to a form of memory recall where some features are recovered faithfully while others are reconstructed based on learned contextual associations.

### 6.2. Consolidation and Replays

All the different forms of memory outlined in the previous section can be learned through the same synaptic update rule given in Equation (Equation 22). In this sense, our model provides a possible unification for most of the known-form of long-term memory in humans and other animals. During training, the only difference between learning a density ϕ(y) or a discrete set of patterns is that in the former case each pattern y is (almost surely) sampled only once while in the latter case each pattern is re-sampled with finite probability throughout training. In generative machine learning, re-sampling is often performed in practice to maximize data efficiency, although it can lead to overfitting since the generative model could memorize the patterns themselves instead of extrapolating the underlying density ϕ(y).

In biological systems, each data point can only be experienced once and this would likely not result in the formation of localized episodic memories, since their encoding requires some form of re-sampling. However, it is well known that the formation of episodic memories in humans requires an extensive phase of consolidation that depends on the activity of the hippocampus. Under our theoretical framework, a possible explanation is that new simulated experiences are generated in the hippocampal network and are then used for synaptic training. In fact, it is well known that these forms of replays can be observed in the hippocampus both during sleep and wakefulness and that their disruption compromises memory consolidation [46]. Training on these self-generated samples can result in a positive feedback, with each reply increasing the probability of the same event being re-sampled in the future. Through this process, a single event can be ‘bootstrapped’ into a self-reinforcing process of activity that can eventually form a localized episodic memory. Furthermore, this would result in the formation of a low-dimensional episodic “core”, as described in the previous sub-section, if only a sub-set of features are re-sampled faithfully, with the other being generated from structural memory.

## 7. Experiments

Figure 2 visualizes the qualitative behavior of both learned and exact score models compared with modern Hopfield iterations. We consider a five-dimensional associative memory problem with four randomly sampled binary patterns. For the forward diffusion dynamics, we used a variance-preserving model as they are numerically more stable and more widely used [23]. We considered both exact score models and trained diffusion models parameterized by three layer perceptrons s(x,t;W) with 750 hidden units in each hidden layer and ReLu activation functions in both hidden layers. The time variable *t* was embedded by appending the value $\theta_{t}=e^{-t/2}$ to the activations of each layer, including the input, hidden, and output layers. This means that each weight matrix acted on a n+1-dimensional feature space, where *n* is the number of activations. We used the training approach described in [23] but with a simpler constant noise schedule (σ = 1). The noise-injection model is given by the equation:(26)$$x\left(t-dt\right)=\left(1-\alpha dt\right)x\left(t\right)+\sigma \sqrt{dt}\delta \left(t\right),$$
where α=0.5 and δ(t)∼N(0,I). Accordingly, the network was trained using the denoiser autoencoder loss:(27)$$L\left(W\right)=E_{\theta_{t}∼U\left(0.6,1\right),\delta ∼N\left(0,I\right)}\;\left[\sum_{n}{∥\delta -s\;\left(\theta_{t}y_{n}+\sqrt{1-\theta_{t}^{2}}\delta ,t;W\right)∥}_{2}^{2}\right].$$

Using this loss, the network is trained to recover the added noise from noise-corrupted patterns. The loss was optimized using Adam with fixed batches including all patterns to be memorized. The networks were trained for 5000 iterations. Diffusion trajectories were integrated using a simple Euler approach for the deterministic ODE samples (see [23]):(28)$$x\left(t+dt\right)=x\left(t\right)+\frac{1}{2}\left(\alpha x\left(t\right)-\frac{s(x\left(t\right),t;W_{\mathrm{trained}})}{\sqrt{1-e^{-2\alpha t}}}\right)dt,$$
while we used a Euler–Maruyama method for the SDE sampler:(29)$$x\left(t+dt\right)=\alpha x\left(t\right)-\frac{s(x\left(t\right),t;W_{\mathrm{trained}})}{\sqrt{1-e^{-2\alpha t}}}dt+\sigma \sqrt{dt}\delta \left(t\right).$$

Both samplers used 300 time steps discretized linearly from $t_{\mathrm{star}}=10^{-3}$ to a *t* corresponding to the noise level. The modern Hopfield iterations were implemented as specified in [26], with β=5 and 4 updates. Consistent with our analysis, all algorithms approximately converge to the same target points. In line with previous work, the modern Hopfield iteration converges after a single iteration, whereas the the diffusion trajectories converge smoothly to the target. However, note that, in this setting, diffusion denoising can also be performed in one step simply by re-scaling the score.

As a first quantitative analysis, we evaluated the Pearson correlation coefficient between the output of modern Hopfield iteration and (a) a diffusion model, (b) a classical Hopfield network, and (c) the ground truth pattern. For a given dimensionality *d*, *n* binary patterns y were randomly generated, and subsequently, corrupted with noise using the formula $\tilde{y}=\theta y+\sqrt{1-\theta^{2}}\epsilon$, where ϵ is a standard Gaussian noise vector. We used a noise level of θ=0.68. We kept the dimensionality equal to 10 and we evaluated the correlation for 10, 20, and 30 stored patterns. The simulation was repeated 100 times in order to reliably compute the correlations. In order to avoid to have to re-train a neural model hundreds of times, for the diffusion models we used the exact score formula (see Appendix A). For the modern Hopfield model, we used 150 updates in order to maximize performance. The final output of all methods was binarized using the sign function. Table 1 shows the estimated Pearson correlations between methods. As expected from our analysis, the correlation between the modern Hopfield iterations and the diffusion model is extremely close to one.

We also performed the same experiments in a completion task, where the patterns were partially zero-masked instead of being corrupted by white noise. The binary masks were sampled randomly from a Bernoulli distribution with p=0.5. Again, the output of the exact diffusion models correlates almost perfectly with the output of the modern Hopfield iterations.

Next, we estimated the error and capacity of the models. The corrupted patterns were fed to the algorithms and the results were compared with the original pattern using the Hamming error. The patterns were considered to be correctly recovered if the error was smaller than 3%. Figure 3a shows the error of an exact diffusion model for different numbers of patterns as function of the dimensionality. For a given noise level and threshold, the capacity was defined as the maximum number of patterns that can on average be recovered. Figure 3b shows the estimated capacity of the exact diffusion model (blue), modern Hopfield network (green), classical Hopfield network (red), and trained diffusion model (black dots). The details of the experiments are given in Appendix A.

## 8. Discussion

In this paper, we demonstrated that a popular class of modern Hopfield networks with exponential non-linearities is mathematically equivalent to a large class of continuous diffusion models at the limit of β→∞. In our experiments, we showed that this equivalence holds almost exactly for a finite β value and for the more stable variance-preserving models, both with exact and trained score. The equivalence depends on the fact that the diffusion models are trained on a finite number of discrete patterns, and in fact the diffusion models can generalize the modern Hopfield energy function to a setting where both episodic memory and semantic memory (i.e., generative manifolds) are jointly encoded by the dynamics of the same network. From the point of view of theoretical neuroscience, this may be used to model the different forms of long-term memory as a result of the same learning mechanism, in a single distributed neural system.

While generative diffusion models offer an attractive paradigm for modeling memory, imagery, and even perception in the brain, more work needs to be done in order for its components to be implemented in a biologically plausible way. In particular, it is unlikely that biological neural networks implement pure noise-injection dynamics. However, the mathematics of generative diffusion models can be written in term of any stochastic differential Equation [23], which can be used to implement more plausible transformation such as, for example, the feedforward perceptual feature as implemented in the hierarchy of sensory cortical areas [47]. More fundamentally, it is unclear how the reverse and forward dynamics can be simultaneously implemented in the brain networks. However, there is tantalizing evidence that this could correspond to the theta rhythm driving periodic cholinergic modulations [37].

The denoising loss given by Equation (Equation 9) is somewhat similar to the self-prediction errors used in predictive coding models [48,49,50]. This opens the door for potentially fruitful connections with existing predictive models of memory [51,52]. Particularly interesting is the recent use of generative predictive coding models for associative memory storage [53], which is also based on a generative model and has some similarities with the approach discussed here.

The main biological issue with episodic memory as conceptualized in this paper is that it seems to require the re-sampling of the same events, while in the real world, each event can only happen once. This can be potentially implemented with some form of bootstrap re-sampling, which might correspond to the replays observed in the hippocampus [20].

## Figures and Tables

**Figure 1 entropy-26-00381-f001:**
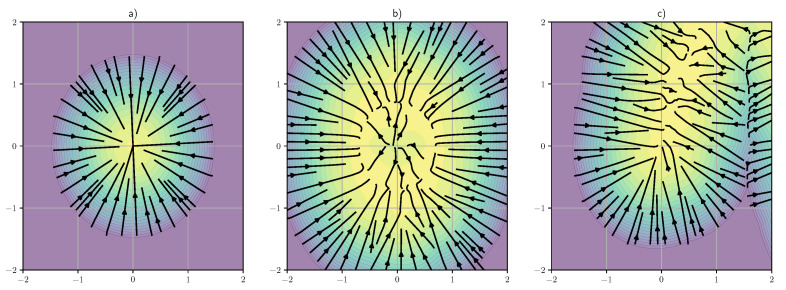
Visualization of different kinds of energy landscape and gradient vector fields corresponding to different forms of memory (in a two-dimensional space): (**a**) classical point-like memory; (**b**) extended localized memory; (**c**) non-localized (semantic) memory structure. The color denotes the probability density of the learned distribution while the lines represent the integral trajectories of the vector field oinduced by the score function.

**Figure 2 entropy-26-00381-f002:**
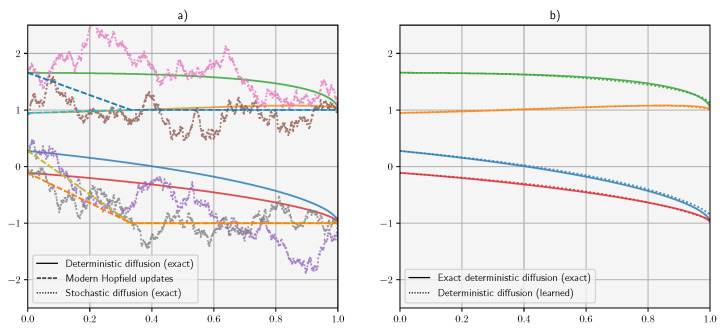
Qualitative analysis of the (marginal) denoising trajectories of a binary associative memory problem with four patterns in a five-dimensional space. (**a**) Comparison between denoising trajectories of diffusion models and modern Hopfield updates. The diffusion curves are integrated using the Euler method with 2000 steps. The trajectories are overlaid to four modern Hopfield updates. (**b**) Comparison between exact and learned deterministic denoising trajectories. The colors are used to identify individual trajectories.

**Figure 3 entropy-26-00381-f003:**
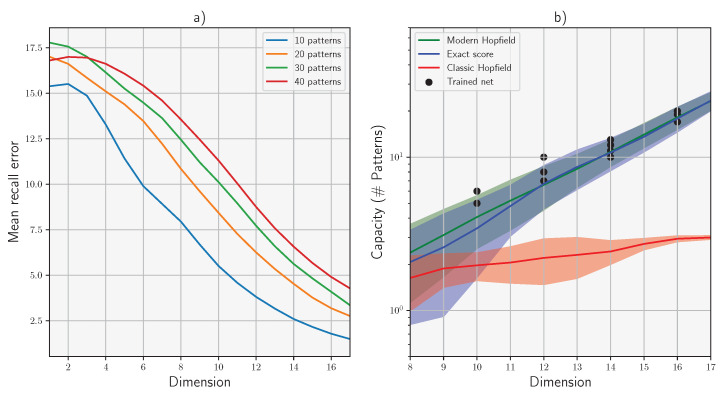
(**a**) Median error of exact diffusion model as function of the dimensionality. (**b**) Capacity of diffusion models and Hopfield networks in log scale. The shaded area denotes the estimated 95% intervals.

**Table 1 entropy-26-00381-t001:** Pearson correlation between output of modern Hopfield network and other models (plus ground truth pattern) in both denoising and completion experiments.

Denoising Task	Number of Patterns	Diffusion Model	Classic Hopfield	True Patterns
	10	0.995	0.732	0.893
	20	0.991	0.704	0.822
	30	0.991	0.715	0.81
Completion task				
	10	0.996	0.741	0.897
	20	0.991	0.707	0.838
	30	0.989	0.700	0.795

Average Pearson correlations between the output of (exact) diffusion models, modern Hopfield networks and classical Hopfield networks in denoising and completion tasks performed on random binary patterns.

## Data Availability

Data are contained within the article.

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
