# Peer review of "In Search of Dispersed Memories: Generative Diffusion Models Are Associative Memory Networks"

_entropy, 2024, doi:10.3390/e26050381_

Round 1
Reviewer 1 Report
Comments and Suggestions for Authors
The author presents a mathematical framework to show the analogies between Modern Hopfield Neural Networks and the Generative Diffusion Model. Even though Hopfield Neural Networks have been used for denoising for many decades the perspective is still quite original. The topic is interesting and the insight the author elaborates from this analogy is extremely relevant to understanding how memory works. The authors also provide potential biological evidence for his interpretation.
Unfortunately, the weakest part of the paper is the mathematical formulation of the framework itself. The author starts with a discussion of the classical Hopfield Neural Network defined on binary variables, and he ends up discussing generative diffusion models defined on continuous variables without presenting new variables and using the Hopfild model notation. The problem is specifically in Eq. 6 where the noise is defined as a real variable and the x is binary.
Moreover, Eq. 7 is claimed to come from ref. 23, but the equation is in nature very different, in part because the noise even if added in steps is considered to be adjustable according to some continuous variable which in our case would be t. Given all this consideration, I am not sure that Eq. 8 holds in this form for discontinuous variables, this should be proven and discussed.
Furthermore, the experimental part should be better described and discussed, the authors give ref. for the reader to study, but the paper should be self-consistent. Also, the author should better explain the implementation details of the three-layer perceptrons with 750 hidden units diffusion models. The info provided does not make the results replicable.
Author Response
Dear reviewer,
We wish to thank you for the detailed review. We appreciate that you found our approach novel and insightful.
1) Continuous vs discrete variables. We are afraid that the reviewer misunderstood some sections of our manuscript. In particular, the theoretical link we discussed is about the continuous modern Hopfield networks as defined in Eq.5. Like generative diffusion models, these newer associative memory models are defined in terms of continuous variables. Similarly, x(t) in Eq.6 is real-valued and not binary. The discrete models in section 1.1 are provided to contextualize the work in the wider Hopfield literature, which was historically focused on discrete variables. We added some clarifications to the text to make clear that our analysis concerns continuous Hopfield models.
Note that, while the model’s variables are continuous, the patterns y can still be binary or discrete.
2) Equation 7 is a standard result of generative diffusion theory, which indeed comes from [23] (among other works). As explained above, both Eq.7 and Eq.8 assume continuous variables, similarly to the variables used in the modern Hopfield energy in Eq.5.
These formulas can be used even when the data (i.e. the patterns) are binary, as the continuous variables will converge to the patterns for t tending to T.
3) We acknowledge that the initial text lacked some important experimental details. Thank you for pointing it out. In the revision, we added more details in section 6. We included the equation forward diffusion model, for the training loss and for the solvers used to integrate the generative dynamics. We also gave complete details about the perceptron architecture and the optimization scheme.
Reviewer 2 Report
Comments and Suggestions for Authors
The manuscript proposes a unified framework and conditions wherein two large classes of long-term memory network models, namely generative diffusion models and modern Hopfield networks, are demonstrated to be equivalent. It illustrates the potential of this framework for capturing certain properties of episodic and semantic memory functions in humans beyond the capabilities of existing associative memory models. This capability facilitates both potential links between the learning processes employed in these two classes of models and new theories concerning how various forms of long-term memory can emerge from a common neuronal network and learning mechanism.
The work is well-reasoned; the manuscript is well-written, organized, and provides detailed descriptions of the methods. The results are clearly presented through both mathematical derivations and simulation experiments. While the manuscript occasionally veers towards overinterpreting the components of the models or learning mechanisms in terms of their correspondence to possible biological structures or processes, it does present a fair discussion of the potential limitations of these interpretations and, consequently, an incomplete understanding of the biological function of long-term memory in the brain.
I would like to particularly highlight one particular interpretation case related to short-term memory (lines 206-217) and how it may account for the feedforward and feedback computations, realized in the loss function of denoising processes underlying memory encoding, as described in the manuscript. Short-term memory is widely understood to arise from interactions among various brain regions, which entail the feedforward encoding of sensory information in sensory cortices and the receipt of feedback signals from downstream areas, such as the prefrontal cortex, which implements a top-down control mechanism.
A recent review paper has described these feedforward and feedback mechanisms using a unified theoretical framework encompassing both discrete spike patterns and continuous oscillation signals for encoding and storing sensory information in sensory areas and their recruitment during memory via feedback interareal communications.
Comeaux P, Clark K, Noudoost B. A recruitment through coherence theory of working memory. Prog Neurobiol. 2023 Sep;228:102491. doi: 10.1016/j.pneurobio.2023.102491. Epub 2023 Jun 29. PMID: 37393039; PMCID: PMC10530428.
That said, the type of feedforward and feedback mechanisms in the context of residuals between noisy states discussed in the manuscript, may not be directly relevant to those known mechanisms in the neuroscience literature, including the recent theory of short-term or working memory mechanisms and circuitry discussed in the aforementioned paper. The authors’ argument on the relationship between short-term memory and model feedback versus feedforward computations needs clarification or discussion in the context of similar biological computations, for example, those reviewed in the paper above or similar studies.
Author Response
Dear reviewer,
We wish to thank you for the detailed and positive review. We are happy that you found the main theoretical and simulation findings of our work to be solid and insightful.
We agree that some of the claims concerning the neurobiological implication should be softened and more grounded in the literature.
Therefore, in the revision we rephrased some claims in section 3 to make clear that it contains speculative ideas. We also included a discussion of more references, including the paper you mentioned. We wish to thank you for drawing our attention to it as it contents several interesting ideas.